

# Unraveling plant-pollinator interactions from a south-west Andean forest in Colombia

Jorge Mario Becoche-Mosquera[1,2,3], Luis German Gomez-Bernal[2,3], Giselle Zambrano-Gonzalez[2,3] and David Angulo-Ortiz[4]

[1] Universidad del Cauca, Popayán, Cauca, Colombia
[2] Ecology and Conservation - GECO, Universidad del Cauca, Popayán, Cauca, Colombia
[3] Universidad del Cauca, Popayán, Cauca, Colombia
[4] Corporación Autónoma Regional del Valle del Cauca, Cali, Valle del Cauca, Colombia

Corresponding author
Jorge Mario Becoche-Mosquera,
jbecoche@unicauca.edu.co

## ABSTRACT

**Background**. Loss of biological connectivity increases the vulnerability of ecological dynamics, thereby affecting processes such as pollination. Therefore, it is important to understand the roles of the actors that participate in these interaction networks. Nonetheless, there is a significant oversight regarding the main actors in the pollination networks within the highly biodiverse forests of Colombia. Hence, the present study aims to evaluate the interaction patterns of a network of potential pollinators that inhabit an Andean Forest in Totoró, Cauca, Colombia.

**Methods**. The interactions between plants and potential pollinators were recorded through direct observation in 10 transects during six field trips conducted over the course of one year. Subsequently, an interaction matrix was developed, and network metrics such as connectance, specialization, nestedness, and asymmetry of interaction strength were evaluated by applying null models. An interpolation/extrapolation curve was calculated in order to assess the representativeness of the sample. Finally, the key species of the network were identified by considering degree (k), centrality, and betweenness centrality.

**Results**. A total of 53 plant species and 52 potential pollinator species (including insects and birds) were recorded, with a sample coverage of 88.5%. Connectance (C = 0.19) and specialization (H2' = 0.19) were low, indicating a generalist network. *Freziera canescens, Gaiadendron punctatum, Persea mutisii, Bombus rubicundus, Heliangelus exortis, Chironomus* sp., and *Metallura tyrianthina* were identified as the key species that contribute to a more cohesive network structure.

**Discussion**. The present study characterized the structure of the plant-pollinator network in a highly diverse Andean forest in Colombia. It is evident that insects are the largest group of pollinators; however, it is interesting to note that birds form a different module that specializes in pollinating a specific group of plants. On the other hand, the diversity and generality of the species found suggest that the network may be robust against chains of extinction. Nevertheless, the presence of certain introduced species, such as *Apis mellifera,* and the rapid changes in vegetation cover may affect the dynamics of this mutualistic network. So, it is imperative to apply restoration and conservation strategies to these ecosystems in order to enhance plant-animal interactions and prevent the loss of taxonomical and functional diversity.

## INTRODUCTION

High Andean forests and moorlands in Colombia are arguably the mountain ecosystems that are most affected by anthropogenic pressures (*Bax & Francesconi, 2019*), which negatively impact the diversity of organisms (*Hazzi et al. 2018*) and ecological processes (*Gonda, 2020*; *Rodríguez-Echeverry & Leiton, 2021*). As a result, ecosystem functions and services, such as pollination, seed dispersal, carbon sequestration, and watershed protection (*Llambíet al., 2019*), are in a highly vulnerable situation. Any alteration to these functions and services could lead to serious problems for the natural environment and the country's economy.

Pollination is a crucial process for maintaining vegetation cover. It involves interactions between angiosperm plants and approximately 300,000 species of pollinating agents, including both vertebrates and invertebrates (*Ollerton, Winfree & Tarrant, 2011*). Pollination is currently affected by pressures on vegetation, such as logging for firewood and timber, agriculture, livestock, and mining (*Goulson et al., 2015*). Additionally, there are threats to pollinators, including declining population sizes, extinction, and an alarming situation for invertebrates (*Nates-Parra, 2016*; *Shivanna, Tandon & Koul, 2020*). The effects that disturbances can have on plant-animal interaction networks are threatening, as they can alter the phenology, distribution, and/or disappearance of their consumers (*Herzog et al., 2010*; *Maglianesi & Jones, 2016*).

In a scenario of global environmental change, it is essential for everyone to learn about the structure, dynamics, and sensitivity of pollination networks and their components in various ecosystems. Research on interaction networks in Colombia is emerging (*Carvajal et al., 2023*; *Ramírez-B et al., 2017*; *Vaca-Uribe et al., 2021*). Several studies focusing on the composition and structure of networks have been carried out in different areas of the country (*Aguado, Gutiérrez-Chacón & Muñoz, 2019*).

There has been limited research on the characteristics of interaction networks in the high Andean mountains, specifically in high Andean forests and moorlands. As a result, there is a scarcity of literature on these networks, making it difficult to determine whether they are similar to or different from ecosystems at lower altitudes. Given the more extreme conditions in terms of temperature, light, radiation, and humidity, it is observed that these networks are less diverse in terms of species and interactions. However, they exhibit greater specialization and modularity. Nevertheless, additional differences may emerge from biogeographic factors, historical utilization of the environment, and global climate change.

Ignorance of the interaction networks allows propose the following questions: (1) What are the components of a plant–pollinator interaction network within the Andean forest? (2) Which interaction patterns may be found in the topology of this network? and (3) What are the key species of the interaction network of this Andean forest?

To address these questions, the present study has the general objective of evaluating the interaction patterns between plants and potential pollinators in a mountain forest located in the southwestern Andes of Colombia. This will be achieved through the following specific objectives: (1) determine the components of the plant–pollinator interaction network in the Andean forest; (2) characterize the interaction patterns within the network's topology; and (3) identify the key species within the interaction network of the mountain forest. Finally, this study aims to address the following hypothesis: if the vegetation in the study area has been modified due to the implementation of productive systems, then the interaction network will exhibit a low specialization index and a generalist structure.

## MATERIALS AND METHODS

### Location of the study area

The study was carried out in El Cofre village, located in the municipality of Totoró, Cauca, Colombia. The field site is located between 2,800 and 3,300 m.a.s.l., at 2°30′50″N, 76°20′14″W, and 2°31′44″N, 76°21′18″W, with an area of approximately seven hectares (Fig. 1). The annual average precipitation is 2,000 mm, with a bimodal weather season pattern. Heavy precipitation occurs during April and May, while low precipitation is observed from November to January (*Martínez, 2011*). The area has a relative humidity between 79.3% and 83.1%. The yearly temperature ranges from 9 to 13 °C. The site has a slope ranging from 20% to 70%. Most of the territory is characterized by mountainous terrain with topography ranging from slightly to extremely rugged (*Arcos, 2009*; *Becoche-M, Macías-P & Zambrano-G, 2018*; *Martínez, 2011*).

The study area is localized within the Guanacas-Purace-Coconuos complex, which is characterized by a vegetation cover classified as Andean Forest, as defined by *Cuatrecasas (1989)*. This area also exhibits significant biodiversity and ecosystem richness, as it is located in a transitional zone between the Andean and moorland biomes. The Cofre River basin, which is a tributary of the Cauca River, plays a crucial role in providing ecosystem services in the area. However, mining activities have led to increased deforestation and have had a significant impact on water contamination (*Arcos, 2009*). As a result, this is one of the last remaining forest remnants in the municipality, and despite the anthropic intervention, it still conserves species that are characteristic of this particular ecosystem.

### Identification and records of network components

During six field trips, 10 transects of 100 m × 2 m were randomly established; each transect was separated from the next by 50 m. The transects were sampled for six months in the year 2019, specifically during the rainy season. All plant–pollinator interactions were recorded during field observations of ornithophilous and entomophilous plants in the blooming phase. Sampling consisted of observing two individuals per plant species from 8:00 to 18:00 during each field trip. Each individual was observed for 30 min, resulting in a total of 360 min, or 6 h per plant species. This approach aimed to obtain the maximum number of interactions (*Vizentin-Bugoni et al., 2016*). Every time a plant was visited, it was marked with visible tape to ensure it could be easily identified and observed during subsequent field trips.

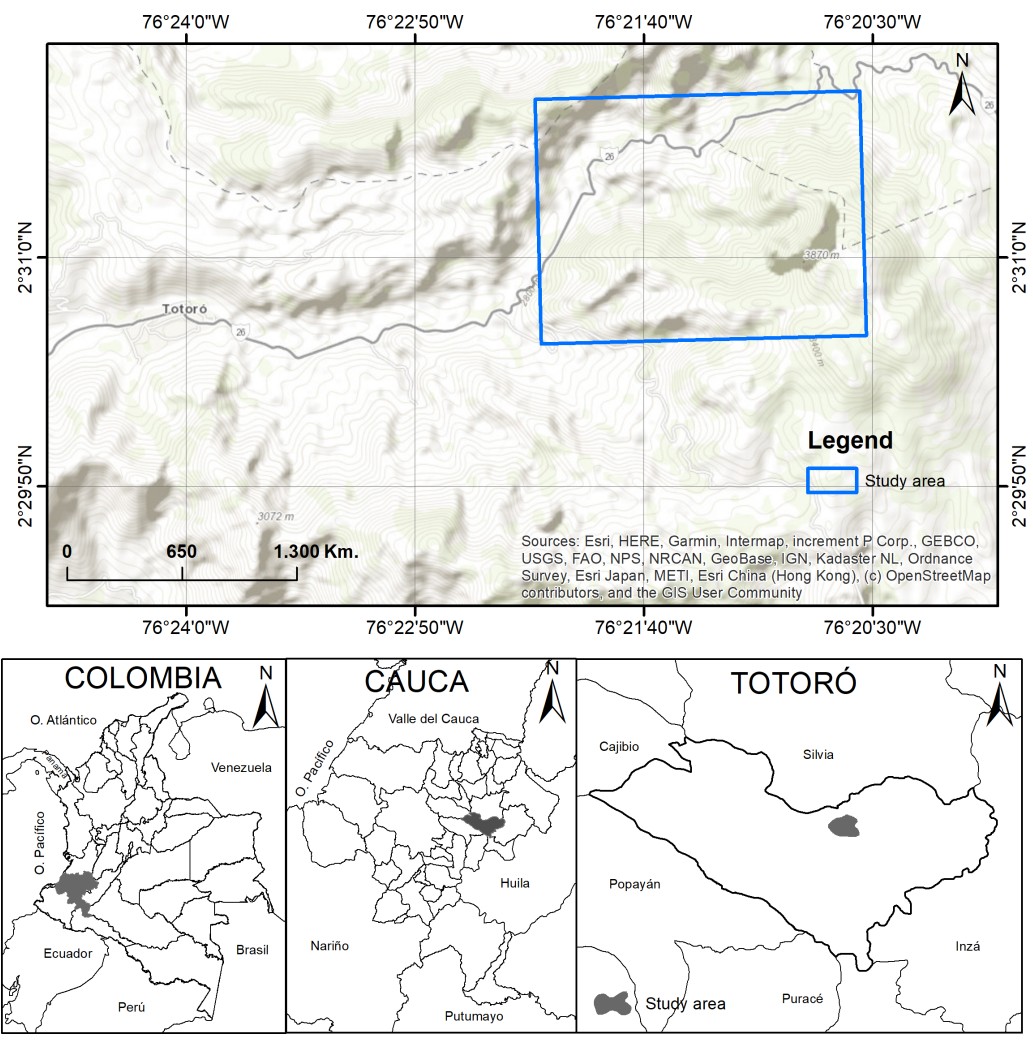

**Figure 1  Study area.** Location of the study area in El Cofre village, Totoró, Cauca. Map credits: Esri, HERE, Garmin, Intermap, increment P Corp., GEBCO, USGS, FAO, NPS, NRCAN, GeoBase, IGN, Kadaster NL, Ordnance Survey, Esri Japan, METI, Esri China (Hong Kong), (c) OpenStreetMap contributors, and the GIS User Community.

Additionally, the observations were supplemented with the visit to each transect in order to evaluate new species of plants in the blooming phase. All the marked plants were visited every other week. The recording of interactions for each marked plant was also randomized during the field trips (*Vizentin-Bugoni et al., 2016*).

The procedure to classify an organism as a potential pollinator consisted of meticulous observation by the researcher, who recorded all the processes taking place in the flower. Three unique behaviors were established to define the potential pollinators: (1) spending a minimum of 3 s on the flower; (2) having pollen on the organism's body; and (3) moving among flowers (*Álvarez & Noval, 2017*).

The vegetation evaluated in the study transects was identified according to the work of *Arcos (2009)*, as were the functional traits of the plants, such as plant habits, height, and

reproduction strategies. When identification in the field was not possible, the samples were collected and taken to the herbarium of the Universidad del Cauca (Cauca State University) to be confirmed by an expert.

Birds were observed and recorded using Bushnell 50x10 binoculars. The classification of the birds was based on the guide by *Ayerbe-Quiñones (2018)*. Unidentified insects were collected using an entomological net and placed in glass vials containing 70% alcohol to be transported to the lab. There, they were identified using the guides by *Borror, Triplehorn & Johnson (1989)*, *González, Ospina & Bennett (2005)*, *Serna (1996)*, *Smith-Pardo & Vélez (2008)* and *Wolf (2006)*, along with some other names confirmed by experts. The specimens will be deposited in the Natural History Museum of the Universidad del Cauca.

The collection of biological material was carried out under Resolution 0152 of February 12, 2015, issued by the National Authority for Environmental Licenses (ANLA), which is the entity responsible for granting permits for the collection of wild species of biological diversity, specifically for non-commercial scientific research purposes.

## Structural parameters of the pollinator network

First, an interaction matrix was developed, where the columns represent animal species (potential pollinators) and the rows represent plants. The frequency of interactions (*i.e.,* the number of visits or *grade*-k) for each species was recorded in this matrix (*Medel, Aizen & Zamora, 2009*). In order to assess the robustness of the sampling, a curve of the cumulative interactions was calculated (*Medel, Aizen & Zamora, 2009*) using an interpolation/extrapolation method. This method was based on the sample coverage, which measures the proportion of the total diversity represented by the recorded data (*Chao & Jost, 2012*).

The connectance, nestedness, specialization, interaction strength asymmetry (ISA), extinction slope, robustness, and modularity of the general network were also calculated. The modularity was determined using the QuaBiMo algorithm (*Dormann & Strauss, 2014*; *Dormann et al., 2022*; *Ramírez-B et al., 2017*).

On the other hand, the present study considered species with higher centrality measurements at the species level as key species in the potential pollinator network structure. The criteria used for this were:

-Species with the highest grade (k), which defines the number of species to which a node is related (*Jordano, 1987*). In animal-plant interaction matrices, the grade indicates the generalization or specialization of each species (*Bascompte & Jordano, 2008*; *Bascompte & Jordano, 2007*).

-Species with higher centrality, which measures the relative proximity to each node in the network structure compared to others. This measure is calculated based on the number and pattern of node connections. High values indicate that the evaluated species is located in the central positions of the network (*Ramírez-B et al., 2017*).

-Species with higher betweenness centrality, which reveals the importance of a node as a connector among different areas of the network. Therefore, the nodes with values higher than zero connect network areas that would otherwise be dispersed or completely

disconnected. It also shows the significance of the species for network cohesiveness (*Newman, 2004*; *Ramírez-B, 2013*).

## Data analysis

All data were calculated using R software version 4.3.0 (*R Core Team, 2023*). The interpolation/extrapolation curves were estimated employing Hill numbers ($q = 0$), sample coverage, and a 95% confidence interval. This was done using the statistical packages iNEXT (*Hsieh, Ma & Chao, 2013*) and SpadeR (*Chao & Jost, 2015*; *Chao et al., 2015*), complemented with the knitr package (*Xie, 2014*). Subsequently, the networks were graphically represented using the Bipartite 2.18 package (*Dormann et al., 2022*; *Dormann et al., 2009*). This package is also used to calculate the original network indices and null models, providing a clear visualization of the network patterns.

In order to evaluate the relevance of metrics, the observed values were compared to those generated through null models (*Dormann et al., 2009*) and thus determined whether the network patterns are natural or if they are generated randomly (*CaraDonna et al., 2017*; *Ponisio, Gaiarsa & Kremen, 2017*). For implementing the null models, the protocol proposed by *García (2013)* and *Gotelli (2000)* was considered. To assess the level of significance, the observed metrics at the network level were compared to 1,000 null models generated using the algorithm developed by *Patefield (1981)*. This algorithm utilizes total fixed peripherals to distribute interactions and produce a series of networks where all species are randomly associated (*Blüthgen et al., 2008*). The significance evaluation was performed using the Bipartite R package (*Dormann et al., 2022*).

## RESULTS

### Network structure of potential pollinators

The potential plant–pollinator network from Totoró Andean Forest is made up of 53 plant species and 52 animal species (hummingbirds -5 sp.-, and insects -47 sp.-) (Appendix 1). A total of 524 interactions out of the 2,756 possible interactions (53 plants × 52 animals) were observed, assuming that all visitors visite all plant species. These were represented in a bipartite network (Fig. 2).

The frequency of visits between animals and plants fluctuated, but there were distinct pairs of species, such as *Gaiadendron punctatum* and *Sphecodes* sp., which, according to the records, had the highest number of visits (Fig. 3). The accumulated interaction curve shows 88.5% representativeness.

The 53 plant species belong to 48 genera of 33 families. The herbaceous layer has the highest number of species (24), followed by shrubs (19) and arboreal species (10). The families with the highest number of recorded interactions were: Asteraceae (109), Melastomataceae (55), Pentaphylacaceae (43), Loranthaceae (41), Lauraceae (39), and Cunoniaceae (39). Genera with the highest documented interactions were *Miconia* (50), *Freziera* (43), *Gaiadendron* (41), *Persea* (39), *Weinmannia* (39), *Gynoxys* (33), and *Baccharis* (29). Although the overall availability of floral resources was not evaluated, it was found that the highest density of flowers was consistent with the botanical families that had the highest number of interactions.

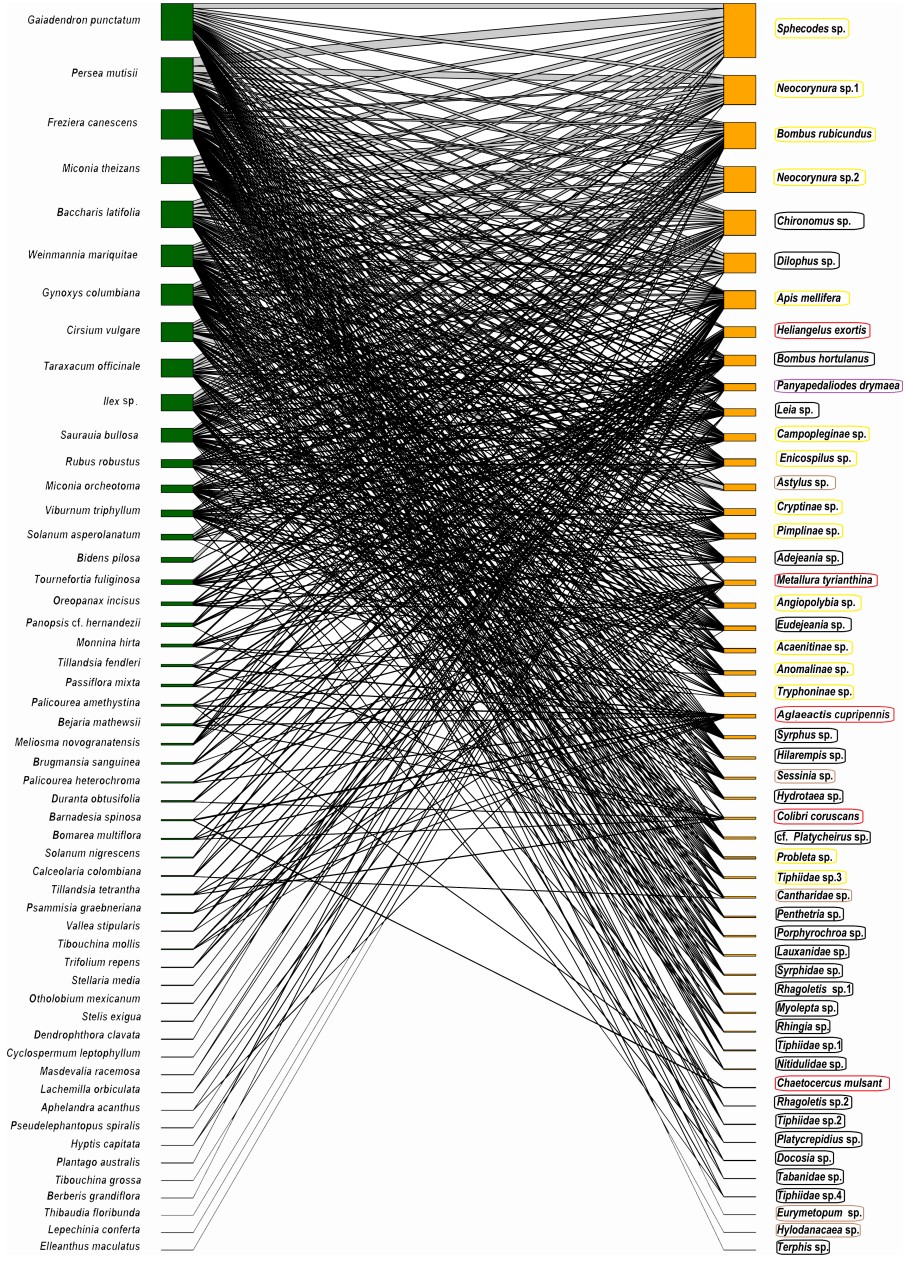

**Figure 2 Network representation of the bipartite interaction between plants (green) and animals (orange).** The width of the interaction shows the level of intensity of the visit. The left column (green color) represents 53 plant species, and the right one represents 47 species of insects (orange color). The five hummingbirds were marked with a red box; the Hymenoptera were marked with a yellow box; the Diptera were marked with a black box; the Coleoptera were marked with a brown box; and the Lepidoptera were marked with a purple box.

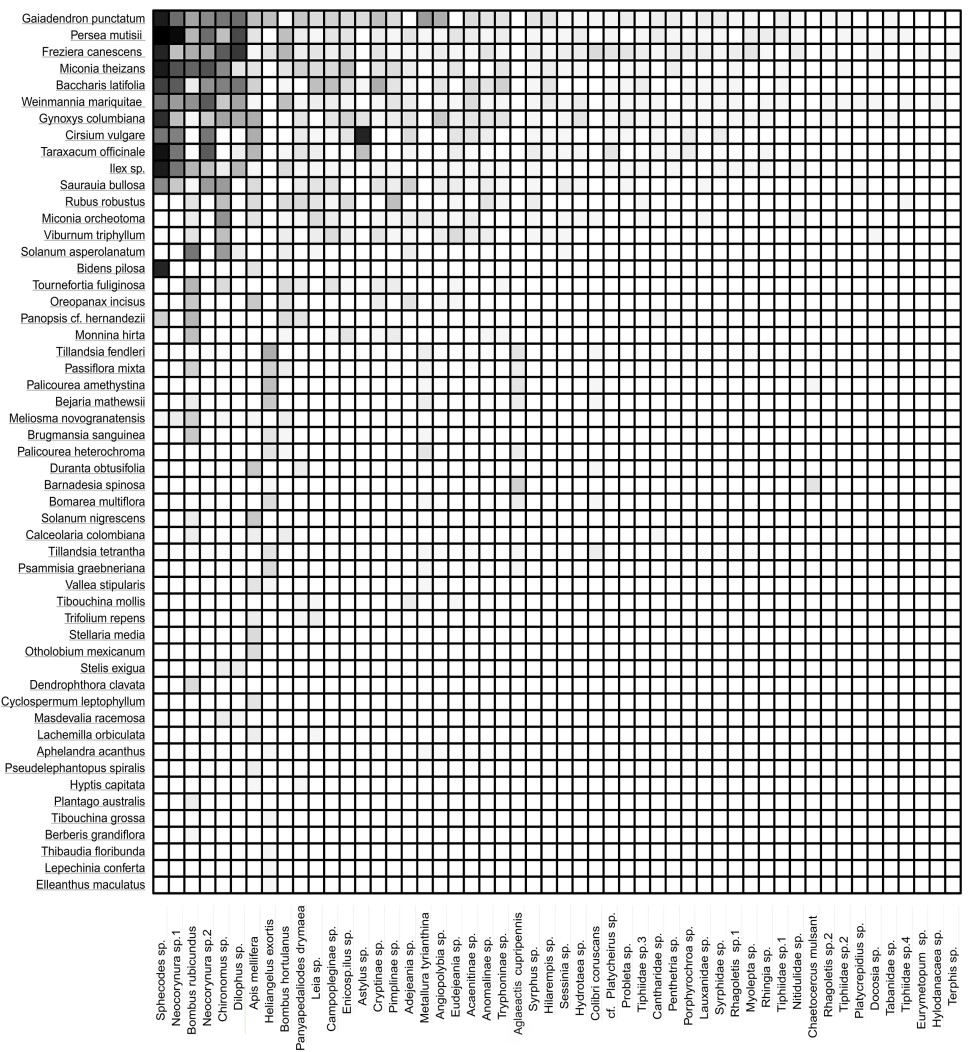

**Figure 3** **Interaction matrix showing the interaction intensity.** The rows in the dataset correspond to the 53 plant species, while the columns represent the 52 animal species. Light color indicates a lower frequency of interaction, while dark color signifies a higher frequency of interaction.

The 52 species of potential pollinators are divided into 47 insect species and five bird species. Insects belong to the orders Hymenoptera (206 unique interactions), Diptera (190), Coleoptera (57), Lepidoptera (18), and 24 families. Among these, Ichneumonidae, Apidae, Halictidae, Syrphidae, and Tachinidae had the highest number of interactions. The only bird family recorded with 53 interactions was the Trochilidae family.

The network had a connectance of 19.1%, indicating a generally low density of interactions between plants and potential pollinators (Table 1). The network specialization (H2′) (Table 1) shows how most potential pollinators, such as *Apis mellifera, Bombus rubicundus*, and *Bombus hortulanus*, interact with the same plant species, including *Freziera canescens, G. punctatum*, and *Persea mutisii*. This indicates a high level of generalization.

**Table 1  Metrics at the network level.** Showing connectance values, H2 (specialization), NODF (nestedness), ISA (interaction strength asymmetry); modularity, gradient of extinction (animals and plants), and robustness (animals and plants) calculated from the original network.

| Metrics | Calculated value |
| --- | --- |
| Connectance | 0.1912192 |
| H2 | 0.1979635 |
| NODF | 55.95717 |
| ISA | −0.02007464 |
| Modularity | 5 |
| Extinction gradient (animals) | 3.5279669 |
| Extintion gradient (plants) | 8.5598719 |
| Robustness (animals) | 0.765515 |
| Robustness (plants) | 0.8750506 |

The nestedness of the network (NODF) value showed a result of 55.95 (Table 1), suggesting that there is a group of generalists in the network of potential pollinators interacting as a sub-group of the species with fewer interactions. The obtained value of the ISA (Table 1) demonstrates that there is a greater dependency of plants on animals than vice versa. It was also found that the network is integrated by well-defined groups of species, as indicated by its modularity (Table 1). It is important to highlight that the five species of birds were part of a different sub-group, separate from the other four sub-groups formed by insects (Fig. 4).

The extinction gradient of the network demonstrates a greater sensitivity to the extinction of potential pollinators when the plants in the system are eliminated. However, the area under the extinction curve suggests that the gradual decrease in the gradient is indicative of a very robust network (sample coverage = 0.885).

Seven key species were identified within the interaction network. These were chosen by observing the highest values of grade, betweenness centrality, and centrality. First, there are three plant species—*F. canescens*, *G. punctatum*, and *P ersea mutisii*—that exhibit the highest values in terms of the number of species they are linked to (grade), their central position in the network (centrality) and their relevance as connectors between different parts of the network (betweenness centrality) (Table 2).

Similarly, it was possible to identify *Bombus rubicundus*, *Heliangelus exortis*, *Chironomus* sp., and *Metallura tyrianthina* as key species in the group of potential pollinators. According to the analysis, these species are the most generalist, as they have the greatest number of links and are also the ones with the closest proximity to each species in relation to the others and greater importance in terms of network cohesion.

Evaluation of metrics of the original network through the *Patefield (1981)* algorithm was significant ($p < 0.001$; H1 accepted) (Table 3), indicating natural patterns of connectance behavior, specialization (H2′), nestedness (NODF), and ISA occurring in the ecosystem. These were determined by looking at the distance between measurements of the recorded values of index value distributions in a random network built by the null model (Fig. 5).

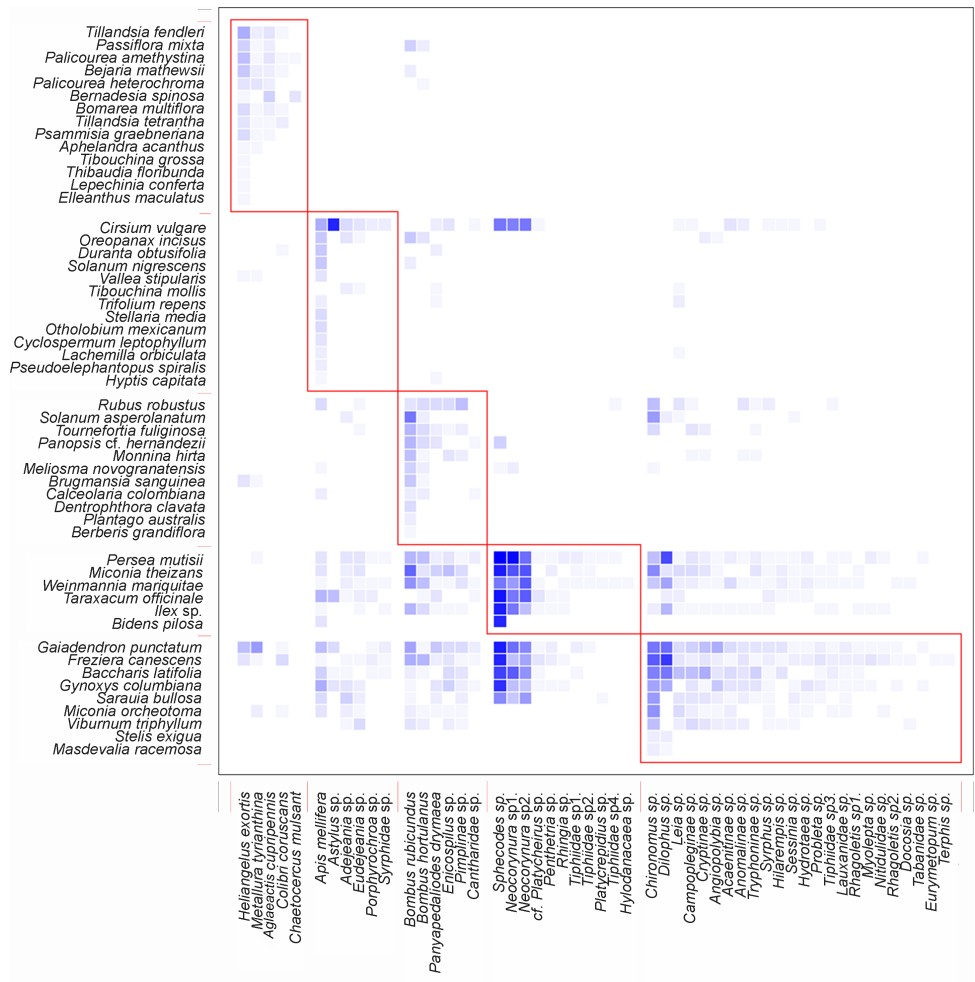

**Figure 4** **Interaction pattern with identified modules for QuaBiMo (steps = 1E8; Q = 0.25)** (*Dormann & Strauss, 2014*). The rows in the dataset correspond to the 53 plant species, while the columns represent the 52 animal species. Darker squares indicate the observed interactions. Red squares outline the five modules. The first module includes the five bird species, while a clear ecological pattern for the other modules is less evident.

## DISCUSSION

The mutualistic network interaction of a cloud Andean Forest in Totoró is a representative case of interactions between plants and potential pollinators living in the area. The obtained sample coverage represents a significant part of the community, in which values such as connectance and network nestedness occur naturally in the ecosystem.

The variations in climatic seasonality, plant phenology, and the wide diversity of insects in the Neotropics (*Basset et al., 2012*) contribute to the challenge of comprehensively documenting all the interactions occurring at the site. During the development of this research, we were able to observe a rainy season during which the number of interactions decreased. This decrease can be attributed to adverse weather conditions, such as wind,

**Table 2 Evaluated metrics for each plant and animal species.** There are 10 plant and animal species with higher calculated values for grade, centrality, and betweenness centrality, making them key species. (*) Key species in the network structure.

|  | **Especie** | **Grade** | **Centrality** | **Betweenness centrality** |
|---|---|---|---|---|
| **PLANTS** | *Freziera canescens** | 43 | 0.14227858 | 0.06285219 |
|  | *Gaiadendron punctatum** | 41 | 0.14227858 | 0.55136541 |
|  | *Persea mutisii** | 40 | 0.07001098 | 0.10273082 |
|  | *Weinmannia mariquitae* | 40 | 0.01944184 | 0 |
|  | *Gynoxys columbiana* | 33 | 0.01740747 | 0.07108799 |
|  | *Baccharis latifolia* | 29 | 0.01944184 | 0.01907239 |
|  | *Ilex* sp. | 29 | 0.01944184 | 0 |
|  | *Miconia theizans* | 28 | 0.01944184 | 0.14217599 |
|  | *Saurauia bullosa* | 23 | 0.01740747 | 0 |
|  | *Miconia orcheotoma* | 22 | 0.07001098 | 0 |
| **ANIMALS** | *Apis mellifera* | 27 | 0.00637904 | 0 |
|  | *Bombus rubicundus** | 26 | 0.08422752 | 0.10849057 |
|  | *Bombus hortulanus* | 21 | 0.08422752 | 0 |
|  | *Panyapedaliodes drymaea* | 18 | 0.00637904 | 0 |
|  | *Heliangelus exortis** | 18 | 0.19077916 | 0.11179245 |
|  | *Leia* sp. | 17 | 0.00637904 | 0 |
|  | *Chironomus* sp.* | 17 | 0.00637904 | 0.125 |
|  | *Eudejeania* sp. | 15 | 0.00637904 | 0 |
|  | *Metallura tyrianthina** | 15 | 0.21460551 | 0.00141509 |
|  | *Campopleginae* sp. | 15 | 0.00637904 | 0 |

**Table 3 Null model to evaluate metrics from the original network.** The expected mean for connectance, specialization (H2'), nestedness (NODF), and interaction strength asymmetry (ISA) resulted from the null model, which is statistically different from the calculated value obtained from the recorded data. The standard deviation (Sd) is calculated from the distribution of random network values generated by the null model, Fisher (F) statistics, and significance value (p). Significance was calculated using Patefield's (1981) algorithm.

| **Original net** | **Observations** | **Null model (Patefield 1981)** | **sd** | **Z** | ***p*** |
|---|---|---|---|---|---|
| **Connectance** | 0.1912192 | 0.2446633 | 0.00303675 | −17.62934 | <0.001 |
| **H2** | 0.1979635 | 0.08281885 | 0.00344194 | 32.7792 | <0.001 |
| **NODF** | 55.95717 | 63.63671 | 1.890555 | −4.103718 | <0.001 |
| **ISA** | −0.02007464 | 0.00453684 | 0.00155516 | −15.99905 | <0.001 |

radiation, and precipitation, which likely hindered the free movement of potential pollinators. The floral phenology could also influence the decrease in interaction records (*Dáttilo & Rico-Gray, 2018*). For example, species like *Miconia orcheotoma* and *Miconia theizans* had flowers throughout all the field trips, while others, such as *Vallea stipularis, Ilex sp., G. pucntatum, Meliosma novogranatensis*, and *Weinmannia mariquitae*, among others, only had flowers during one or two field trips (*Martínez, 2011*; *Ospina, 2009*).

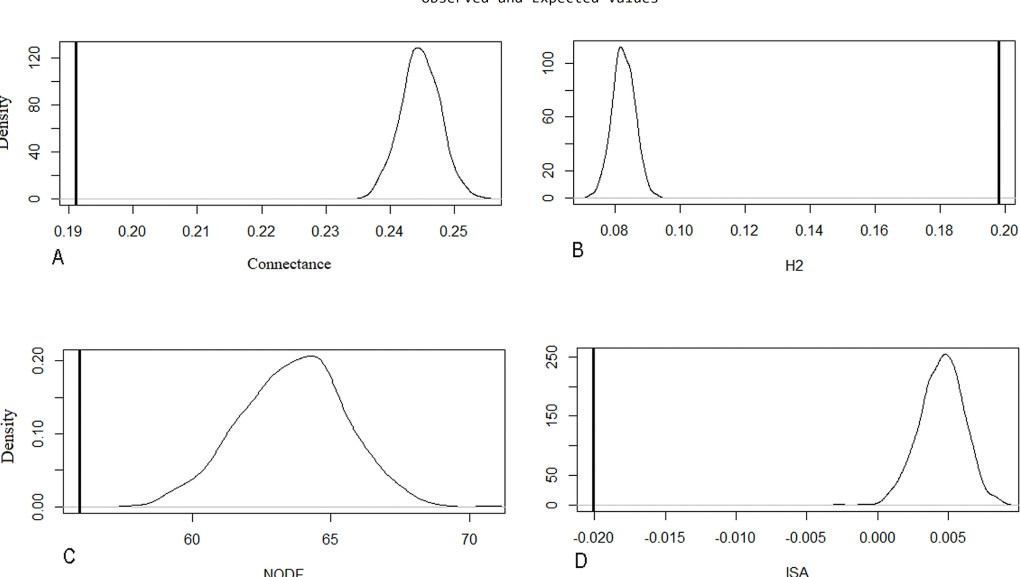

**Figure 5** **Null model graphic for evaluated metrics on the original network.** The $X$-axis represents the index value, and the $Y$-axis indicates the density of the model for each index unit. A line representing the observed value is compared to a bell curve representing the distribution values of a random network built by the null model. The distance between these two measurements is significantly different. The following parameters are represented: (A) connectance, (B) specialization (H2′), (C) nestedness (NODF), and (D) interaction strength asymmetry (ISA).

However, the low number of observed interactions compared to those that were possible can also be explained by the presence of forbidden links in the network (*Ramírez-B, 2013*). These forbidden links may be due to differences in size between the androecium and the pollinator or between the size of the beak and the corolla. An example of this is the absence of a registry for hymenopteran or dipteran insects that are less than one centimeter long and interact with large flowers ($\geq 10$ cm), such as *Brugmansia sanguinea* or *Passiflora mixta*. However, authors like *Vizentin-Bugoni, Maruyama & Sazima (2014)* mention the importance of these types of interactions for the structure of ecological networks, which may be more significant than species abundance.

Some of the interactions between pairs of species were recorded with greater intensity compared to others. Most of these interactions happened with generalist species of insects, such as *Sphecodes* sp., *Neocorynura* sp1., *Bombus rubicundus*, *Neocorynura* sp2., and plants such as *G. punctatum*, *Persea mutisii*, and *F. canescens*. The intensity of such interactions might be linked to an ecological strategy adopted by potential pollinators, mostly insects, which consists of maintaining fluctuating and abundant populations as well as a high number of offspring (*Begon, Townsend & Harper, 2006*). That dominance of these organisms also contribute to a higher frequency of interactions (*Bascompte et al., 2003*; *Vázquez et al., 2009*).

The plants documented in the network structures are characterized by their small size, small leaves, and a canopy filled with abundant epiphytes. These characteristics are

representative of this particular life zone (*Cuatrecasas, 1989*; *Rangel-Ch., 2015*). In addition, the wide variety of species in the herbaceous layer indicates that the forest is in the early stages of regeneration. Pioneer species from the Melastomataceae and Asteraceae families are dominant (*Arcos, 2009*; *Vargas, 2008*), and these families had the highest number of observed interactions. It is important to highlight the role of such pollinators during the early succession stages. Some studies have demonstrated that these organisms are more abundant in regenerating ecosystems than in mature forests, primarily because of the greater availability of resources (*Roberts, King & Milam, 2017*; *Tauro, 2013*). Pioneer plants from families such as Asteraceae and Melastomataceae, which exhibit high phenotypical plasticity and colonization capacity, provide attractive flowers and rewards for pollinators (*Vargas, 2008*). In this study, pioneer species such as *Baccharis latifolia* and *Miconia theizans* belonging to these families, respectively, were recorded.

Additionally, it has been found that pollinator diversity is similar in both recovering ecosystems and conserved ones. This suggests that these organisms play a role in re-establishing biological connectivity (Roberts). Plants with zoophile pollination has been proposed as an ecological attribute when choosing potential species for regeneration, as they may aid in accelerating the regeneration process in ecological dynamics (*Vargas, 2008*).

As for the potential pollinators, birds and insects were identified. The observed hummingbirds (*Chaetocercus mulsant, Aglaeactis cupripennis, H. exortis, Metallura etallura tyrianthina,* and *Colibri coruscans*) belong to the studied territory (*Ayerbe-Quiñones et al., 2008*), and their interactions occurred only with bush, arboreal, and vine plants, such as *Duranta obtusifolia, F. canescens,* or *Passiflora mixta.*

However, most of the recorded pollinators were insects. This could be due to the presence of a high density of typical floral resources in plant succession processes. In addition, this forest fragment is surrounded by an agricultural matrix, which can result in a higher frequency of certain insects (*Becoche-M, Macías-P & Zambrano-G, 2018*). The Hymenoptera order is characterized by a wide variety of pollinator species, including bees and wasps. This can be easily demonstrated by the interactions observed during field trips, where species such as *Bombus rubicundus, Bombus hortulanus,* and *Angiopolybia* sp. were frequently seen. Without a doubt, this order is one of the major contributors to the ecosystem service of pollination (*Nates-Parra, 2016*). The Diptera order also presents a wide range of interactions; however, *Borror, Triplehorn & Johnson (1989)* mention that the number of species devoted to pollination is lower compared to Hymenoptera. It is also important to emphasize that some researchers, such as *Dalsgaard et al. (2009)* and *Devoto, Medan & Montaldo (2005)*, point out how the richness and composition of pollinator species vary according to elevation, precipitation, and temperature. Additionally, these authors also mention a global pattern where dipteran richness is predominant in humid areas while hymenopteran richness is predominant in dry land. Although the present research was carried out in a humid territory, an equal number of species were found for both orders. This is probably attributed to factors such as a wide variety of pioneer plant species that dominate the early succession stage of the forest. These plants offer floral appeals and rewards that are utilized by hymenopteran and dipteran insects. On the other

hand, there could be other species whose habitat has changed as a result of environmental phenomena, such as climate change (*Kiers et al., 2010*; *Michener, 2007*; *Tylianakis et al., 2008*). Nevertheless, understanding ecological patterns in the tropics is challenging due to the limited research conducted in these areas.

Likewise, the network metrics were calculated, but they cannot be compared at a quantitative level to other networks due to differences in size and unknown distributions of each recorded variable (*Ramírez-B, 2013*). Nonetheless, below is a descriptive analysis of the interaction patterns.

As for network connectance, its low value is attributed to the wide diversity of recorded species and interactions. Some authors suggest that this happens because when species richness increases, the number of interactions also increases, but connectance decreases exponentially (*Jordano, 1987*; *Ramírez-B, 2013*; *Winemiller, 1989*). Research studies, such as those conducted by *Basilio et al. (2006)* and *Olesen et al. (2008)*, have identified similar patterns in these parameter behaviors.

Network specialization shows that most potential pollinators tend to be generalists, although some species with only a single interaction can be found, such as *Terphis* sp., which is extremely unusual, with distribution and behavior still unknown. However, specialization is often the result of insufficient sampling. However, when this is evaluated over several years, most of the recorded organisms tend to be generalists (*Petanidou et al., 2008*). It is important to highlight that specialized pollinators undergo co-evolutionary processes that are being seriously threatened by activities that degrade ecosystem health. This poses a risk of extinction for species that rely on these pollinators for sexual reproduction and cannot replace this dependency (*Lindberg & Olesen, 2001*).

The recorded nestedness value can be attributed to interactions between abundant and dominant species with a high number of interactions, such as *F. canescens* and *W. mariquitae*, and potential pollinators, such as *Eurymetopum* sp. and *Hylodanacaea* sp., that were recorded visiting the former species only a few times. The nestedness pattern may suggest heterogeneity in the evolutionary development of each species, thereby generating greater selective effects among the species that interact more frequently (*Medel, Aizen & Zamora, 2009*). Yet, some authors, like *Rico-Gray (2007)*, emphasize the importance of obtaining a substantial sampling effort to replicate the methodology in neighboring areas. This is necessary in order to affirm the co-evolutionary patterns that are recorded in the pollination networks.

The interaction asymmetry index suggests that potential pollinators in this network have a greater dependence on plants than the opposite. This can be attributed in part to the evolutionary strategy adopted by the majority of angiosperms, wherein energy expenditure can be minimized while achieving high reproductive success through interactions with pollinators (*Nates-Parra, 2016*). Likewise, *Vázquez et al. (2007)* suggest that the high relative abundance of some species can significantly influence the increase in asymmetry in interaction strength. This finding is consistent with the results of our research, which identified potential pollinators such as *Sphecodes* sp., *Bombus rubicundus*, *Neocorynura* sp1., and *Chironomus* sp. that were recorded in high abundance in the area, probably due to the agricultural and livestock activities in the surrounding study area. However, it

would be important to further evaluate the role of species such as *Sphecodes* sp., which are generally kleptoparasitic bees without highly developed scopes, why their contribution to the pollination process may not be as effective as in other species (*Nates-Parra, 2016*).

Network modularity indicates that there are five compartments. One of these compartments comprises five bird species, while the other four compartments consist of insects. Even though there were insect interactions with ornithophile plants, a sub-group formed between hummingbirds and bell-shaped flowers. This is the result of co-evolutionary processes that allow plants to reproduce in exchange for a reward (*Medel, Aizen & Zamora, 2009*). The specialization of the members within this module is the main reason for their existence (*Dormann & Strauss, 2014*).

The remaining four compartments were formed by insects and the rest of the plant species, but as mentioned by *Dormann & Strauss (2014)* in their research, a possible ecological pattern is less evident. In order to ecologically interpret these modules, it is necessary to have a deep understanding of them. It is crucial to identify and conserve the species that serve as hubs and links among modules, as their removal from the network can trigger cascades of extinction (*Olesen et al., 2007*). On the other hand, it is also possible that new modules may be generated after documenting all interactions, including potential pollinators such as bats and/or moths, when analyzing the data.

Other evaluated metrics included the extinction slopes, which indicated the high sensitivity of potential pollinators to extinction if plants are removed from the network. According to *Memmott, Waser & Price (2004)*, this is a result of the decline in various specialist organisms, which tend to disappear first. This finding aligns with the information found in the present research. Nonetheless, there is a high value of robustness for animals and plants, which increases the network's tolerance to extinction. This is attributed to network redundancy (*Kaiser-Bunbury et al., 2010*; *Memmott, Waser & Price, 2004*). This behavior was also recorded in the present study, where most plants interacted with many pollinators.

Seven key species were identified based on their contribution to the network structure. The first species recorded were *F. canescens, G. punctatum*, and *Persea mutisii*. This species is essential for the forest structure, as they have higher horizontal and vertical dominances compared to other species (*Arcos, 2009*). They may also be plants that offer significant flower rewards (pollen and nectar), which guarantees their reproduction and interaction with numerous pollinators. According to *Tinoco et al. (2016)* *G. punctatum* produces an average of 0.09 ml of nectar and has a sugar concentration of 31 mg/ml, which is high compared to other evaluated plants. On the other hand, these species can contribute to ecological restoration processes, strengthening the resilience and function of the network (*Kaiser-Bunbury et al., 2017*).

*Bombus rubicundus* and *Chironomus* sp. were also identified as representing the hymenopteran and dipteran orders that stand out for being generalists and abundant in the study area. Among them, *Bombus rubicundus* is considered an important pollinator of the Andean forest in the country (*González, Ospina & Bennett, 2005*).

*H. exortis* and *Metallura tyrianthina* are hummingbird species characteristic of this life zone area, observed in the field with greater frequency compared to the other observed bird

species. Added to this, the number of interactions and the frequency of visits are important factors in maintaining network cohesiveness. This is supported by studies conducted by *Bascompte et al. (2003)* and *Martín-González, Dalsgaard & Olesen (2010)*; that highlight the importance of generalist species in maintaining network cohesiveness. They explain that the highly heterogeneous distribution of interactions among species provides alternative pathways to find answers when the system is disturbed.

It is remarkable to observe the presence of each taxonomic group (plants, birds, and insects) as key species, highlighting the essential role that all network components play in the dynamics of the ecosystem. It is also important to emphasize that even though pollination networks are dynamic (*CaraDonna et al., 2017*) and tolerant of the extinction of their components (*Memmott, Waser & Price, 2004*), the elimination of key species, as identified in this study, can lead to the secondary extinction of other potential pollinators and plant species.

*Apis mellifera* was also identified as an important pollinator of the native vegetation in the study area. However, it is important to note that this species was introduced from Europe (*Michener, 2007*) and has successfully adapted to the different ecosystems in the country. Even though the ecosystem services offered by this species, such as pollination and derivative products, are important, studies have shown that the presence of this species displaces native pollinators. This displacement leads to a decrease in the production of fruit and seeds from the native flora (*Montero-Castaño et al., 2018*; *Valido, Rodríguez-Rodríguez & Jordano, 2014*). The impact caused by an invasive species on the ecosystem can result in new adaptation mechanisms in native plants. These mechanisms may include changes in flowering stages, flower morphology, and flower reward, causing a loss in the diversity and functionality of the pollinator network (*Pisanty & Mandelik, 2012*). Although other studies also mention that these particular species only play a secondary role, affecting the native species mildly (*Flórez-Gómez, Maldonado-Cepeda & Ospina-Torres, 2020*).

On the other hand, evaluating the structural parameters of the original network using a null model revealed that connectance, specialization, nestedness, and ISA are inherent patterns in the ecosystem. However, it is important to understand that pollination networks are dynamic over time and that their interactions are not static. This is because communities can undergo structural rewiring as new interactions are established among the remaining species in response to changing environmental conditions (*CaraDonna et al., 2017*; *Costa et al., 2018*). This suggests that, despite the loss of species, the network components would rearrange their interactions (*CaraDonna et al., 2017*), mostly because of the niche width of the persistent species and the reduction of intra-specific competition (*Medel, Aizen & Zamora, 2009*).

Nevertheless, the continuous elimination of species from the system due to diverse anthropic or climatic factors could lead to the disappearance of fewer generalist nodes that have specific interactions with the initially extinct ones, as suggested by *González et al. (2009)* and *Herzog et al. (2010)*. This usually occurs because many species cannot establish new interactions within the community when there is a rewiring in the network (*Brodie et al., 2014*). Consequently, this results in a potential loss of species and functional diversity.

## CONCLUSION

In this research, potential pollinators and their interaction patterns with vegetation were identified in a cloud forest in the Colombian Andes. Insects and birds make up groups of pollinators with varying degrees of specialization and sensitivity to environmental changes. The conformation of different modules suggests that there are specific relationships between birds and groups of plants, indicating a certain degree of specialization within the network. However, the predominant structure persists among generalists, primarily due to the dominance of species commonly found in altered systems. Additionally, differences in the processes of resilience and extinction of key species could impact taxonomic and functional diversity, community stability, and the provision of ecosystem services. Therefore, it is necessary to continue researching and monitoring the network. Moreover, the information obtained is useful for designing conservation strategies that aim to restore and conserve the high mountain ecosystem. This includes considering the identified key species and articulating research with environmental education and community management.

## ACKNOWLEDGEMENTS

The authors would like to express their gratitude to: Mr. Giovanni Varona Balcázar for granting permission to the researchers to conduct their research in his natural reserve, and Professor Nelsy Elvira Benitez Luna for her invaluable support in proofreading the manuscript for style and grammar.

### Funding

The translation of this manuscript was financed (501100005682) by the Master's Program in Biology from the Universidad del Cauca. This work was supported by the Natural History Museum of the Universidad del Cauca and the general administration of the Universidad del Cauca. The funders had no role in study design, data collection and analysis, decision to publish, or preparation of the manuscript.

### Grant Disclosures

The following grant information was disclosed by the authors:
The Master's Program in Biology from the Universidad del Cauca: 501100005682.
The Natural History Museum of the Universidad del Cauca.
The general administration of the Universidad del Cauca.

### Competing Interests

David Angulo-Ortiz is employed by Corporación Autónoma Regional del Valle del Cauca. The others authors declare that they have no competing interests.

### Author Contributions

- Jorge Mario Becoche-Mosquera conceived and designed the experiments, performed the experiments, analyzed the data, prepared figures and/or tables, authored or reviewed drafts of the article, and approved the final draft.

- Luis German Gomez-Bernal conceived and designed the experiments, performed the experiments, analyzed the data, prepared figures and/or tables, authored or reviewed drafts of the article, and approved the final draft.
- Giselle Zambrano-Gonzalez conceived and designed the experiments, performed the experiments, analyzed the data, prepared figures and/or tables, authored or reviewed drafts of the article, and approved the final draft.
- David Angulo-Ortiz conceived and designed the experiments, performed the experiments, analyzed the data, prepared figures and/or tables, authored or reviewed drafts of the article, and approved the final draft.

## Animal Ethics

The following information was supplied relating to ethical approvals (i.e., approving body and any reference numbers):

The collection of biological material was carried out under Resolution 0152 of February 12, 2015 issued by the National Authority for Environmental Licenses—ANLA for which a framework permit for the collection of wild species of biological diversity was granted for non-commercial scientific research purposes (Resolution 0152 of February 12, 2015—ANLA).

## Field Study Permissions

The following information was supplied relating to field study approvals (i.e., approving body and any reference numbers):

The collection of biological material was carried out under Resolution 0152 of February 12, 2015 issued by the National Authority for Environmental Licenses—ANLA for which a framework permit for the collection of wild species of biological diversity was granted for non-commercial scientific research purposes (Resolution 0152 of February 12, 2015—ANLA).

## Data Availability

The raw data, including the matrix where all interactions were recorded, is available in the Supplemental File.

## Supplemental Information

Supplemental information for this article can be found online at http://dx.doi.org/10.7717/peerj.16133#supplemental-information.

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
