# Peer review of "Unraveling plant-pollinator interactions from a south-west Andean forest in Colombia"

_PeerJ, doi:10.7717/peerj.16133_

## Round 0.1 · original submission · Major Revisions

Dear Authors: Please revise the paper for clarity in presentation. Figures and tables could be reduced. A clear hypothesis and objectives should be outlined. Several areas are unclear. Please edit for language and grammar. Editing by a fluent English speaker is recommended.

·

Basic reporting

The manuscript represents a good effort at evaluating mutualistic interactions in megadiverse regions. However, it still requires work to communicate a story to the reader.

Figures and tables are ok in general but are too many, so authors are encouraged to present only what is essential to their story in a way that is digestible to their desired audience.

Experimental design

The original primary research is within the aims and scope of the journal.

Authors need to do a better job at stating their research questions and tailoring the story they want to communicate. They have enough results to shape a nice story.

Methods are well described but need additional details in terms of things they measured (or not) and how that is relevant to their results.

Validity of the findings

Authors present novel results, despite offering too many things that are not easy to digest.
All underlying data have been provided; they are robust, statistically sound, & controlled.

The discussion and conclusion need to be better tailored to a more succinct story told by the authors.

Additional comments

I congratulate the authors for conducting this work, which is a difficult task in Colombian forests. Determination of insect identities is particularly challenging in a region with scarce local key for identification, so I think it’s even brave to conduct this type of work. However, I think you still need to give more shape to this manuscript. Please think about the audience that you think is interested in this work, and reflect on the three main messages you’d like to convey to this audience. After that, please try to recheck your manuscript so that it is more succinct and direct in terms of the ideas you want to communicate.

It’ s sad that most Q1 journals in the natural sciences only take English to conduct the peer review process. Until this changes, I encourage authors to use public tools like Grammarly and others to check for suggestions on grammar structure and appropriateness. This includes a special invitation to check for appropriate plural vs singular forms (species and “species” and others – heading of descriptions for all of your figures), anglicisms found in the document (pollinizers vs pollinators), punctuation signs ( using “,” vs “;” with no clear intention – lines 208-211; 224: 225), word choice in expressions such as “ecological duties” (line 357), use of “&”, “and”, and “y” when presenting references in the main text (line 144), and overall writing structure. These corrections take some time but make it easier for the reader to understand all that you are trying to share.

In general, I’d suggest rechecking the flow of the narrative in your paper because sometimes it is difficult to understand the message that you are trying to convey. You can recheck paragraphs so that they conform to a general structure in which the first line presents the general idea of the paragraph, and then there is a development of that idea with your evidence or that obtained from other sources. Also, you can check if some of the things you present are essential for the work that you conduct, and tailor the narrative accordingly. For example, many of your paragraphs at the introduction are true but do not necessarily support the case of your work (ie. talking about food security and pollinators). I suggest taking a paper reporting mutualistic interactions and seeing the structure of their introduction to get an idea of what may be more relevant to include in your own case.

Introduction
Recheck the relevance and flow of your paragraphs for introducing the case. I’d stress the fact that Colombia tends to be a black box or huge gap for this type of studies, and the area you sampled may be in a biodiversity hotspot (if so, I’d try to describe why and how that is relevant for the study).

Please (re)state the study questions that you develop in the study.

Methods
I’m curious: You report an altitudinal frame of between 2800-3000 masl. In other regions of the country, this would be a frame of transition between Andean forests and Paramo, and marked changes in terms of the vertical structure of the vegetation and overall composition of plant communities. How is this different from Totoró?

Did you evaluate the overall availability of floral resources (abundance)? And how that could influence the structure of the interaction matrix and what you call key species? That is not clear from the document.

Results:
Recheck that you present results according to your study questions.
You have nine graphs. Please reconsider which are essential for developing your questions and which could be part of the supplementing material of the paper.
In figures 5: 7, there are no labels “Y” for the Y axis. Please rectify.


Discussion:
The discussion is very long and sometimes it does not seem to be discussing your results. Please try to see if you are discussing ideas for which you present results. For example, lines 310-318 talk about seasonality and phenology, but in your case, that is only presented in the methods. Please make sure you tailor your discussions to the results you present. Also, please check for the structure of your paragraphs. Make sure they are complete paragraphs developing the idea you present in the first sentence.

What do you mean by “Most of the potential pollinizers were insects, and this could be a consequence of a wide micro habitats variety and of optimal ecological conditions for their survival”? it’s very unclear. The first part of the sentence does not seem to be explained by the second part (the second part could easily be applied to any other biological group). Please clarify.

I’d advise checking for a recent paper on mutualistic interactions conducted by Nathalia Florez, Prof. Rodulfo Ospina, and other colleagues; and for the work of Prof. Marisol Amaya. It may be useful.

Reviewer 2 ·

Basic reporting

The language is not always clear, and could benefit from greater English proficiency

Not all relevant literature is cited appropriately, and insufficient field background/context provided

The article structure, figures and tables are professional and raw data are shared.

Self-contained. Authors could expand on hypotheses and predictions at end of introduction.

Experimental design

Original primary research within aims and scope of journal.

Research question generally well defined, though authors could state more clearly context of what is known/not known in the context of the tropics.

In general it seems like this study is a rigorous investigation performed to a high technical & ethical standard.

Insufficient details are provided in the methods to fully ascertain the robustness of the sampling, and thus greater expansion of methods is required.

Validity of the findings

Data have been provided.

Conclusions are generally well stated and linked to original question.

Additional comments

In this study the authors describe the plant-pollinator network from an Andean forest in Totoró Colombia. This is a highly biodiverse region of the world that has been historically under-studied and thus this work is an important contribution.

My major general comments:
1) Information is missing in the methods regarding the sampling protocol, and the sampling dates, which limits the readers ability to discern whether the design/sampling is sufficient. In the discussion authors mention the rainy season, but from the methods it is impossible to know when the authors surveyed the network (and if each plant species was surveyed both in the rainy/dry season, or if the survey of different species is confounded by date). Please provide more details and greater clarity on sampling protocol.

2) The introduction and discussion would greatly benefit from further context on the stark contrast between the high levels of biodiversity in Colombia (among the highest in the world) with the lack of knowledge in peer-reviewed publications regarding the functioning of many ecosystems in the country, especially in certain regions. Discussing what is and is not known in the context of Colombia and the tropics more broadly (both neo-tropics and beyond) would be an important addition.

3) Authors are using terms interchangeably in many instances and need to standardize for consistency and clarity (e.g. pollinizer/pollinize/etc. should be pollinator/pollinate/etc., the authors use connectance and connectivity interchangeably, but in the context of this manuscript I believe it should always be connectance, “bonding/bonds” should be “interacting/interactions”, etc.). I recommend the authors re-read their manuscript with this in mind and standardize terminology as much as possible.

4) The R code is not provided, nor do the authors have a data availability statement.

General comments:

• L22 Add information briefly on sampling (for how long, in how many places and when)?
• L55 Nates-Parra is not the correct citation. Change to Ollerton, Winfree and Tarrant 2011 Oikos (https://doi.org/10.1111/j.1600-0706.2010.18644.x)
• L59 Similarly changes Nates-Parra to Klein et al. 2007 PRSB (https://doi.org/10.1098/rspb.2006.3721)
• L64 refer to can refer to habitat loss, pesticide use, pathogens, climate change and other stressors and cite Goulson et al. 2015 DOI: 10.1126/science.1255957
• L84 when discussing palynological libraries in Colombia cite Obregon & Nates-Parra 2014 https://doi.org/10.1007/s13744-013-0186-5
• L104 citation missing for deforestation and water contamination sentence
• L115 how did the researchers ensure that they surveyed all/most of the plants in the area? Transect surveys? If so, how were those completed? "random routes" is insufficient detail
• L 115 how large is the area being sampled? Insufficient details on sampling are provided
• L116 "It was deemed to collect the same
• number of individuals by each plant species (2 samples), and standardized the timing for sampling for every worked day (8:00-18:00h)" is very unclear. Are you saying that you observed the visitors/potential pollinators arriving at two different individuals of each plant species? When was each visit? Give dates and additional information.
• L116 were all species sampled at the same time, or was the sampling randomized within the 8:00-18:00h sampling range? Did that vary between the two sampling events per species?
• L119 can you add here the number of total plant species sampled too (e.g. "... for each plant species (x plant species surveyed, for a total of x hours of observation total)
• L123-124: what does "detailed of the forage" and "following movements to the flowers" mean?
• L 151 I do not know what the authors mean by this sentence. Please clarify entire section on k.
• L165 this seems like an old version of R, and it seems like the number is actually referring to R not to the version of R studio. Ensure that the version of R provided is up to date.
• L183 Were all non-insect visitors hummingbirds? If so, state as much.
• Because of "forbidden links", not all of those are possible. Could you say "potential ones if all visitors were able to visit all plant species" instead?
• L194 Do authors have permission to reproduce the bird illustrations? If so, state as such, if not, remove.
• L234-238 this sentence is unclear. Maybe to clarify specifically state that the first refer to plants and that the second group refers to pollinators?
• L233 what is visible bonding? connectance? As mentioned above, it is important to standardize terminology.
• L206 In the methods authors never described how they made the distinction of the functional traits of each plant, add to the methods
• L276 “According to field records” is unclear, clarify what you mean.
• L285 Table 3 doesn’t have a figure legend here like the others do
• L311 There is no mention of season in the methods and the results. If the authors studied over multiple seasons, include that in the analyses and include a figure showing differences/or lack thereof of interactions in rainy vs dry season
• L368 also include that much of the historical research has focused on temperate areas, and we are much more limited in our understanding of ecological patterns in the tropics, including in Colombia
• L382 Add citation and discussion of Petanidou et al. 2008 paper showing that specialization is often a product of insufficient sampling (when evaluated over years most are generalist)
• L398 missing a section on life history of the dominant plant and pollinators found. For example, Sphecodes are kleptoparasitic bees, which have different foraging strategies than bees foraging for pollen provisioning
• L448 add Page & Williams, 2023, Ecology (https://doi.org/10.1002/ecy.3939)
• Figure 1 insufficient contrast, could the authors change the color of the area of interest to be blue or another color so it can more easily be distinguished?
• Figure 2 Can the authors add a symbol to represent the different orders (hymenoptera, diptera, etc.)? Could be done by adding a figure legend and having asterix or other symbols next to the general groups of visitors for readers to more clearly be able to see general patterns
• Figure 2 The plant names have a space while the pollinator names have a period, it would be best if they all had a space between the genus and the species, and if both were italicized
• Figure 2 make this plot include interactions, and then move Figure 3 to the supplementary materials
• Figure 4 Add a typical rarefaction curve (sample coverage on y axis based on number of interactions on the x axis) as plot B of figure 4, and refer to in methods, results and discussion (to see if it approaches an asymptote). For plot A clarify legend: Because rarefaction curves are expected to reach an asymptote, this title is confusing to the reader. Change to "Coverage-based rarefaction (solid line) and extrapolation (dashed line) plots for interaction richness on Hill numbers q = 0. Confidence interval is equal to 95%" (I am assuming 95% percent, but please update as necessary
• Figure 5 – 7 add figure legends
• Figure 6: color bars by family (then can move Figure 5 plot to supplementary materials). Sort by number of interactions on x axis instead of alphabetical
• Figure 8 add the same legend recommended above to the pollinator names (e.g. if asterix represents hymenoptera it could be something like * Apis.mellifera) so it is easier to see the composition of the groups (especially since one of your results is that one of the modules is primarily birds
• Figure 9 move to the supplementary materials
• Table 3 what what the threshold for "highest metric values"? Unclear
• Tables chance <0.00 to <0.001 (values are not less than zero)


Recommendations for greater clarity in the text
• L18 replace "the country forests'' " with "Colomba's highly biodiverse forests"
• L19 "the goal, to analyze" to "aims to analyze"
• L 19 "structural dynamics" is unclear --> evaluate interaction patterns is more clear
• L25 can delete ", and with the help of a Bipartite package”
• L26 methods -> models
• L29 potential pollinizer species among species and bird --> "potential pollinator species (insects and birds)
• L35 "Through present study it was achieve the characterization of the function and structure of complex mutualism interactions" --> The present study characterized the structure of the plant-pollinator network in a highly diverse Andean forest of Colombia. That would more clearly articulate what the work did
• L38 "is a robust one, against extinction chains" --> "may be robust against extinction chains"
• L 42 void --> avoid
• L52 conservancy --> conservation
• L63 problematic -> problem
• L120 specie  species
• L123 pollen into the body on the organism --> pollen on the body of the organism
• L137 delete “place”
• L145 What percent of taxa were identified to species vs genus?
• L150 add weighted and unweighted degree as well, since that is one of the most commonly used metrics in the literature, which is also one of the most straightforward to understand
• L162 o --> or
• Results
• L183 change “among” to “namely”
• L183 add number of each species after hummingbirds and insects
• L206 change "33 are families and 48 are genera", to "from 48 genera in 33 families"
• L250 reference the sample coverage figure here
• L254 NOD -> NODF
• L289 change "can be proven" to "was determined"
• L346 change "due to they may" to "because they may"
• L405 change vegetable to plant
• L406 suggested edit for clarity, ", a possible ecological pattern is less evident, and deeper knowledge about the system and each of its components is required to be able to interpret such modules"
• L409 falls of extinction --> extinction cascades
• L410 motts -> moths
• L461 o -> or
• L463 change “due to” to “because”

---

## Round 0.2 · Minor Revisions

Please revise based on reviewer comments. Make sure you provide a line by line response.

Reviewer 2 ·

Basic reporting

Clear English, relevant literature cited, professional figures/tables, raw data shared, self-contained with relevant results to hypotheses.

Experimental design

Research questions well defined, methods sufficiently well described.

Validity of the findings

Data are provided and they contribute to our understanding of the diversity of plants and pollinators in the high Andean mountain, a biodiverse and understudied region.

Additional comments

The authors did not provide a line-by-line response to previous reviewer comments so it was hard to assess which comments they responded to, and which ones they did not (and when they didn’t, the justification for not doing so). Having gone back through my previous review and the new version of the manuscript, I can see that the authors have addressed many of my previous comment and that the resulting manuscript is much clearer. I would have liked to see justification for why certain suggested edits were not incorporated. In addition, I include additional minor revision below, with the lines corresponding to the PeerJ reviewing PDF document.

Minor:
Abstract
L16 delete the “the” before “ecological”
L17 delete the “,” before “such as”
L19 replace “networking” with “networks”
L30 replace “evidencing” with “indicating”
L35 replace “pollination” with “pollinators”
L37 this sentence is not entirely clear. One possible rephrasing for clarity is “On the other hand, the diversity and generality of the species found suggests that the network may be robust against chains of extinction”
Introduction
L45 replace “probably” with “arguably”
L45 add the citation to support the claim that
L61 can also add Vaca-Uribe et al. 2021 https://doi.org/10.1111/afe.12460
L71 add “Historical” before the “Disregard”
L71 replace “can be” with “are”
L87 delete “, was found”
L96 replace “each transect was distant from the other 50 m” with “each transects was separated from the next by 50 m”
L120 where were the insects ultimately deposited?
L133 cite the bipartite package here (as that is what I am assuming you used to calculate the different network metrics); you mention in line 154 that the package was used to visualize and later in line 163 to calculate the null models, but I assume you also used it to calculate the original network indices as well? Please clarify.
L166 potential is written twice
L166 replace “network pollinators” with “plant-pollinator network” as you mention both the plants and the potential pollinators later on in the sentence
L168 what is the 53 x 52?
L176 add a space after the period
L196 replace “The metrics at the network level were calculated: a connectance of 19.1% can be observed” with “The network had a 19.1% connectance”
L199 and 200 “and” shouldn’t be in italics
L261 replace “to obtain” with “with obtaining”
L264 replace “impeded” with “likely impeding”
L277 add a space after the :
L294 it would be interesting to note that Asteraceae
L284 replace “characterize for” with “are characterized by”
L306 replace “relict” with “fragment”
L308 replace “characterizes for” with “is characterized by”

---

## Round 0.3 · Minor Revisions

1. The conclusion needs more work. Add the following points to the conclusion section:
- Discussion some restoration or conservation options based on your results
- Summarize the caveats of this research and propose opportunities for future research
2. Language edits: Please check your manuscript for language and grammar edits with the help of a fluent English speaker or some Editing service. This will help in improving the quality of the presentation.
3. While the aim is presented at the end of the introduction section, specific objectives and hypotheses are missing. Please add specific objectives clearly listed. This should be followed by hypotheses that are being tested in this study.

**Language Note:** The Academic Editor has identified that the English language must be improved. PeerJ can provide language editing services - please contact us at [email protected] for pricing (be sure to provide your manuscript number and title). Alternatively, you should make your own arrangements to improve the language quality and provide details in your response letter. – PeerJ Staff

---

## Round 0.4 · accepted · Accept

The authors have addressed the editor's comments.